# Learning Time-Varying Convexifications of Multiple Fairness Measures

## Abstract

There is an increasing appreciation that one may need to consider multiple measures of fairness, e.g., considering multiple group and individual fairness notions. The relative weights of the fairness regularisers are *a priori* unknown, may be time varying, and need to be learned on the fly. We consider the learning of time-varying convexifications of multiple fairness measures with limited graph-structured feedback.

## 1 Introduction

Artificial intelligence has gained widespread popularity and adoption across diverse industries due to its ability of automatic decision-making processes. In numerous contexts where artificial intelligence permeates various aspects of our lives, from business operations to societal dynamics and policy formulation, ensuring fairness is of greatest importance to meeting environmental, social, and governance standards. While for nearly any problem in the field of artificial intelligence, there can exist multiple measures of individual fairness as well as multiple measures of subgroup fairness. Often, Subgroup fairness involves multiple protected attributes (e.g., race, sex), creating numerous combinations of subgroups and corresponding subgroup fairness measures, all of which deserve consideration. Hence, it becomes essential to take into account the trade-offs among optimising for multiple fairness measures. There have been designs of systems to balance multiple potentially conflicting fairness measures (Awasthi et al., 2020; Kim et al., 2020; Lohia et al., 2019).

In the long term, the societal preferences and definitions of fairness are subject to continuous evolution and changes. There have been concerns that naively imposing fairness measures to decision-making policies can harm minorities as delay effects (Liu et al., 2018c; D'Amour et al., 2020). Maintaining artificial intelligence models adaptable to dynamic fairness concepts, due to evolving data distributions, or changing societal norms, has been considered (Bechavod & Roth, 2023; Wen et al., 2021; Zhang & Liu, 2021; Jabbari et al., 2017) in sequential decision algorithms, using feedback or effects to decision-making policies.

Consider, for example, political advertising on the Internet. In some jurisdictions, the current and proposed regulations of political advertising suggest that the aggregate space-time available to each political party should be equalised in the spirit of "equal opportunity" [1]. It is not clear, however, whether the opportunity should be construed in terms of budgets, average price per ad, views, "reach" of the ad, and whether it should correspond to the share of the popular vote in past elections (e.g., in a previous election with some means of data imputation), the current estimates of voting preferences (cf. rulings in the US allowing for the participation of only the two leading parties), or be uniform across all registered parties (cf. the "equal time" rule in broadcast media). See Koppel (1983); Miller (2013); Hiltunen (2021) for standard references and Martinez (2018); MacCarthy (2020); Jaursch (2020) for an informal discussion. For example, on Facebook, the advertisers need to declare their affiliation with the political party they support[2]. Considering multiple political parties, as per the declaration, and their budgets, one may need to consider multiple fairness measures (e.g., differences in average price per ad, differences in spend proportional to the vote share, differences

---

[1] In the USA, see the equal opportunity section (315) of the Communications Act of 1934, as amended many times.

[2] In the USA, this may become a legal requirement, cf. the Honest Ads Act. Cf. `https://www.congress.gov/bill/115th-congress/senate-bill/1989`

in the reach, etc, and their $l_1, l_2, l_\infty$ norms). Furthermore, most platforms have a very clear measure of ad revenue, which they wish to balance with some "fairness regularisers" Lu et al. (2020). To complicate matters further still, the perceptions of the ideal trade-off among the ad revenue and the fairness regularisers clearly change over time.

Initially, we try to model fair sequential decision algorithms by incorporating feedback from all kinds of fairness regularisers, which however is generally limited in practice. As mentioned earlier, fairness measures are not independent from each other: some measures may conflict with each other, while others may align to some extent. Decisions made in each time step may enhance one fairness measure but have a negative impact on another. We regard such relation as a (un)directed graph with nodes being fairness regularisers and available actions, and edges being relation among these nodes. We consider the edges being time-invariant (in Case I), and time-varying (in Case II) to model the situations of every-changing social norms of fairness. The works of graph-structured bandits (Mannor & Shamir, 2011) allow us to integrate the graph representing relationships between fairness regularisers and actions into the process of fair sequential decision-making.

Our work is inspired by Awasthi et al. (2020) which utilise an incompatibility graph among multiple fairness regularisers (criteria) and assume that full feedback from all fairness regularisers is available. Our method consider limited graph-structured feedback with the graph presenting relation among fairness regularisers and actions.

## 2 RELATED WORK

### 2.1 FRAMEWORKS FOR REASONING ABOUT MULTIPLE FAIRNESS MEASURES

Very recently, there have been several attempts at formulating frameworks for reasoning about multiple fairness measures. The fairness resolution model, proposed in Awasthi et al. (2020), is guided by the unfairness complaints received by the system and it can be a more practical way to maintain both group and individual fairness. This work provides a finite-state discrete-time Markov chains framework that supports multiple fairness measures and takes into account their potential incompatibilities. It leads, however, to PSPACE-Complete decision problems, which are hard to solve in practice, independent of whether P equals NP. Independently, Ospina et al. (2021) proposed a framework that utilises an online algorithm for time-varying networked system optimisation, with the aim to trade-off human preferences. Particularly, the function of human preference (fairness measure function) are learned concurrently with the execution, using shape-constrained Gaussian processes.

Alternatively, one could consider approaches from multi-objective optimisation (MOO), where key recent references include Zhang et al. (2021) who consider dynamic MOO, but do not provide any guarantees on the performance of the algorithms. This, in turn, is based on a long history of work on convexifications (Sun et al., 2001, e.g.) in MOO. There is also related work across many applications of MOO, such as in Mathematical Finance Li & Ng (2000).

### 2.2 GRAPH-STRUCTURED BANDITS

Once we formulate our framework, we present algorithms applicable therein, which draw upon the work on online non-smooth convex optimisation Zinkevich (2003). We refer to Cesa-Bianchi & Lugosi (2006); Shalev-Shwartz et al. (2011); Hazan (2016) for recent book-length introductions. In the case of graph-structured feedback, there are numerous algorithms summarised in Table 1, which are based on the tradition of online convex optimisation with bandit-feedback Kleinberg (2004); Agarwal et al. (2010); Bubeck et al. (2015); Hazan & Li (2016); Ito (2020). We discuss these in more detail later and in the Supplementary Material.

## 3 OUR FRAMEWORK FOR REASONING WITH MULTIPLE FAIRNESS REGULARISERS

As the need for additional notions of model fairness, and the establishment of a trade-off among them increases, it is necessary to build up a comprehensive framework for reasoning about multiple fairness regularisers, especially when not all regularisers can be fully satisfied simultaneously. Let

| Algorithm | Reference | Regret (allowing for minor variations) |
|---|---|---|
| ELP | Mannor & Shamir (2011) | |
| ExpBan | Mannor & Shamir (2011) | $\mathcal{O}(\sqrt{\overline{\chi}(G)\log(k)T})$ |
| Exp3-Set | Alon et al. (2013) | $\tilde{\mathcal{O}}(\sqrt{\alpha T \log d})$ |
| UCB-N | Caron et al. (2012) | Expected regret parametrised by clique covers |
| Exp3-Dom | Alon et al. (2013) | |
| Exp3-IX | Kocák et al. (2014) | Stochastic feedback graph |
| UCB-LP | Buccapatnam et al. (2014) | Stochastic feedback graph |
| Exp3.G | Alon et al. (2015) | Tight bounds for some cases |
| Exp3-WIX | Kocák et al. (2016) | $\tilde{\mathcal{O}}(\sqrt{\alpha^* T})$ |
| Algorithm 1 | Cohen et al. (2016) | Tight bounds for some cases matching Alon et al. (2015) |
| BARE | Carpentier & Valko (2016) | |
| Exp3-DOM | Alon et al. (2017) | $\mathcal{O}\left(\log(K)\sqrt{\log(KT)\sum_{t\in[T]}\mathrm{mas}(\mathcal{G}_t)}\right)$ |
| ELP.P | Alon et al. (2017) | $\mathcal{O}\left(\sqrt{\log(K/\delta)\sum_{t\in[T]}\mathrm{mas}(\mathcal{G}_t)}\right)$ w.p. $1-\delta$ |
| TS | Tossou et al. (2017) | |
| TS | Liu et al. (2018b) | Optimal bayesian regret bounds |
| OMD | Arora et al. (2019) | Switching costs |
| TS+UCB | Lykouris et al. (2020) | |
| IDS | Liu et al. (2018a) | |
| UCB-NE | Hu et al. (2020) | Non-directed graphs |
| UCB-DSG | Cortes et al. (2020) | Pseudo-regret bounds |
| | Li et al. (2020) | Cascades in the stochastic f.g. |
| OSMDE | Chen et al. (2021) | $\mathcal{O}((\delta^* \log(|V|))^{1/3} T^{2/3})$ |
| | Lu et al. (2021) | Adversarial corruptions |

Table 1: An overview of the algorithms for the model with graph-structured feedback. $\mathrm{mas}(\mathcal{G}_t)$ is the size of the maximal acyclic graph in $\mathcal{G}_t$, $\chi$ is the colouring number, $\delta^*$ is the weak domination number, $\alpha$ is the independence number, $\alpha^*$ is the effective independence number.

us consider discretised time and $T$ be the number of time steps (rounds). Let $[T] = 1, \ldots, T$. The set of vertices includes action vertices $V_a = \{a^1, \ldots, a^I\}$ and fairness regulariser vertices $V_f = \{f^1, \ldots, f^J\}$. Further, we assume there exists a possibly time-varying, (un)directed compatibility graph $\mathcal{G}^t = (V_a, V_f, E^t)$, where edges $E^t$ represent the (possibly) time varying relation among the multiple actions and fairness regularisers vertices $V_a, V_f$.

We follow Alon et al. (2015) to define the neighbourhood of action vertices. A directed edge from an action vertex $a^i$ to a regulariser vertex $f^j$ represents that conducting action $a^i$ will affect regulariser $f^j$. This relation is captured by in-neighbours $\mathrm{N}_{\mathrm{in}}^t(f^j)$ of a regulariser vertex $f^j$, in equation 1, defined as a subset of action vertices which could affect $f^j$ in round $t$. On the other hand, a directed edge from an action vertex $a^{i1}$ to another action vertex $a^{i2}$ represents that conducting action $a^{i1}$ will disclose the reward if conducting action $a^{i2}$. This relation is captured by out-neighbours $\mathrm{N}_{\mathrm{out}}^t(a^i)$ of an action vertex $a^i$, in equation 1, defined as a subset of action vertices whose reward will be disclosed if we conduct $a^i$ in round $t$.

$$\mathrm{N}_{\mathrm{out}}^t(a^i) := \{a^n | a^n \subseteq V_a, (a^i \to a^n) \in E^t\}, \tag{1}$$

$$\mathrm{N}_{\mathrm{in}}^t(f^j) := \{a^n | a^n \subseteq V_a, (a^n \to f^j) \in E^t\}. \tag{2}$$

Each regulariser vertex $f^j$ is characterised by a state $s^{(j,t)} \in \mathcal{S}$ in round $t$. The state $s^{(j,t)}$ evolves if any in-neighbours of regulariser vertex $f^j$ are conducted, as in equation 3:

$$s^{(j,t)} = P\left(s^{(j,t-1)}; a^{(i,t)}, i \in \mathrm{N}_{\mathrm{in}}^t(f^j)\right), \tag{3}$$

where $P$ is the state evolution function and $a^{(i,t)} \in \{0,1\}$ is the value of action vertex $a^i \in \mathcal{A} \subset \mathbb{R}_+$ in round $t$. The values of action vertices are chosen with $\sum_{i\in[I]} a^{i,t} \leq 1$, for all $t \in [T]$. The

reward $r^t$ is comprised of revenue and a concave combination of weighted rewards from regularisers. Overall,

$$r^t\left(a^{(i,t)}, i \in [I]\right) := c^t\left(s^{(i,t)}, \ldots, s^{(J,t)}\right) + \sum_{j \in [J]} w^{(j,t)} f^j\left(s^{(j,t)}\right),\qquad(4)$$

where $c^t(\cdot)$ denotes performance criterion, e.g., revenue, as a function of states of regularisers. The fairness reward from the regulariser $f^j$ in round $t$ is a function of its state $s^{(j,t)}$, denoted $f^k(s^{(j,t)})$, with weights $w^{(j,t)}$. Note that $c^t, w^{(j,t)}$ could be time-invariant or time-varying. Please refer to the Supplementary Materials for a table of notation. Within this general framework, we consider:

- **Limited graph-structured feedback:** While we do not know $c^t, w^{(j,t)}, P$, we are given the compatibility graph $\mathcal{G}^t$ in the beginning of each round, where $\mathcal{G}^t$ could be time-invariant or time-variant. We decide $\{a^{(i,t)}\}_{i \in [I]}$ and then observe some rewards associated with action vertices in $\mathrm{N}_{\mathrm{out}}^t(a^i)$, where $a^{(i,t)} > 0$, according to $\mathcal{G}^t$. This limited feedback is utilised for the next round.

Notice that in the limited-feedback case, Awasthi et al. (2020) introduced a compatibility graph of regulariser vertices only in the fairness context. It considers a sequence of fairness reward, as a function of regulariser vertices' states, received at each time step, which reflect users' perceptions of fairness regarding outcomes, and is a function of its current state. On the other hand, we could also include the real fairness outcomes as feedback because there might be some distance between users' perceptions and reality.

Suppose Algorithm A is used to decide $\{a^{(i,t)}\}_{i \in [I]}$ in each round, we have the overall reward $R(A) := \sum_{t \in [T]} r^t\left(a^{(i,t)}, i \in [I]\right)$. We consider two types of regrets:

- **Dynamic regret:** the difference between the cumulative reward of A and that of the best sequence of actions $\{k^{(i,t)}\}_{i \in [I], t \in [T]}$, $k^{(i,t)} \in \{0, 1\}$ chosen in hindsight, thus $\mathrm{OPT}_D - R(A)$, where

$$\mathrm{OPT}_D = \max_{\sum |k^{(i,t)}| \leq 1} \sum_{t \in [T]} r^t\left(k^{(i,t)}, i \in [I]\right).\qquad(5)$$

- **Weak regret:** the difference between the cumulative reward of A and that of the best single action $\{k^{(i)}\}_{i \in [I]}$, $k^{(i)} \in \in \{0, 1\}$ chosen in hindsight, applied at all time steps, thus $\mathrm{OPT}_W - R(A)$, where

$$\mathrm{OPT}_W = \max_{\sum |k^{(i)}| \leq 1} \sum_{t \in [T]} r^t\left(k^{(i)}, i \in [I]\right).\qquad(6)$$

## 4 MOTIVATING EXAMPLES

Let us revisit the example of political advertising on the Internet, which we mentioned in the introduction. Let us consider two major political parties, e.g., Conservative and Liberal parties, and several third-party candidates in a jurisdiction, where political advertisements on social networking sites are regulated.

Figure 1(a) give a general example of the compatibility graph, with 3 action vertices, 3 regulariser vertices and 9 subgroup regulariser vertices. Particularly, according to definitions of edges, action vertex $a^1$ could potentially affect $\{f^1, f_1^1, f_1^2, f_1^3\}$ and the rewards associated with $a^2, a^3$ would be disclosed if we conduct $a^1$. It could model the situation where 3 regularisers of political advertising $f^1, f^2, f^3$ required dollar spent, reach, number of shares to be equalised respectively ("equal opportunity"). Those subgroup regularisers vertices $f_k^1, f_k^2, f_k^3$ required dollar spent, reach, number of shares of party $k$ to be within certain ranges, respectively. There might be a lot more actions but here we consider $a^1, a^2, a^3$ as selling one unit of advertising space-time to Conservative party, Liberal party, and third-party candidates.

As another example, let us consider the same action vertices ($I = 3$) but three new regulariser vertices ($J = 3$) whose states $s^{(j,t)}$ is a representation of how much of the advertising space-time (e.g.,

the number of shares) has been allocated to the political party $j$ until round $t$. we suppose that the initial states of these regulariser vertices are $\{s^{(j,0)}\}_{j\in[J]} = [0,0,0]$. If the regularisers suggest the allocation $\{s^{(j,t)}\}_{j\in[J]}$ to be proportional to the the share of the popular vote, one may set regularisers $f^1, f^2, f^3$ to be requiring the two major parties and all third-party candidates to get an equalised allocation of political ads on a social media platform. That is, $s^{(j,t)} / \sum_{j\in[J]} s^{(j,t)} = 1/J$ for $j \in [J]$. Then in each round, each regulariser $f^j$ returns a penalty, i.e., the minus reward, for the allocation of the associated party $j$ being away from its target share, such that $f^j(s^{(j,t)}) = -\left| s^{(j,t)} - \frac{s^{(j,t)}}{\sum_{j\in[J]} s^{(j,t)}} \right|$.

The revenue of the platform is assumed to be $c^t\left(s^{(i,t)}, \ldots, s^{(J,t)}\right) := (C^t)' s^{([J],t)}$, where $C^t \in \mathbb{R}^J$ is the revenue vector and $(C^t)'$ is the transpose of $(C^t)$. Let us consider $T = 2$, and the revenue vector $C^t$ for the two rounds to be:

$$C^1 = [0 \quad 1 \quad 0], C^2 = [1 \quad 0 \quad 0]. \tag{7}$$

Here, we would like to learn the trade-off between the regularisers and the revenue of the platform, only with the single scalar weight $w^{(j,t)} = 0.1$. The value of action vertices are binary, i.e. $a^{(i,t)} \in \{0,1\}$, with $\sum_{i\in[I]} a^{(i,t)} \leq 1$, which depending on the discretisation of time, and may be single advertisements or all advertisements auctioned off within a given time slot. Correspondingly, the state transmission function $P$ is set to be $s^{(j,t)} = s^{(j,t-1)} + a^{(j,t)}$.

**OPT$_D$:** In the example above, the best sequence of actions would choose a different $a^i, i \in I$ within the budget $B = 1$ of each round. Perhaps it could conduct $a^1$ in the first round and $a^2$ in the second round, with the corresponding actions and states being equation 8.

$$\begin{aligned}
\{a^{(i,1)}\}_{i\in[I]} = [0 \quad 1 \quad 0], \{a^{(i,2)}\}_{i\in[I]} = [1 \quad 0 \quad 0], \\
\{s^{(j,1)}\}_{j\in[J]} = [0 \quad 1 \quad 0], \{s^{(j,2)}\}_{j\in[J]} = [1 \quad 1 \quad 0].
\end{aligned} \tag{8}$$

Let $s^{([J],t)}$ be the $J$-dimensional matrix $[s^{(1,t)}, \ldots, s^{(J,t)}]'$. The resulting reward is

$$\begin{aligned}
\text{OPT}_D &= \sum_{t\in[T]} (C^t)' s^{([J],t)} - 0.1 \sum_{k\in[K]} f^j\left(s^{(j,t)}\right) \\
&= 2 - 0.1 \times \left( \left| 1 - \frac{1}{3} \right| + 2 \times \left| 1 - \frac{2}{3} \right| + 2 \times \frac{1}{3} + \frac{2}{3} \right).
\end{aligned}$$

**OPT$_W$:** If, on the other hand, we were to pick only a single action vertex to be taken in both rounds $t$, that is $a^{(i,1)} = a^{(i,2)}$. The corresponding actions and states would be:

$$\begin{aligned}
\{a^{(i,1)}\}_{i\in[I]} = \{a^{(i,2)}\}_{i\in[I]} = [1 \quad 0 \quad 0], \\
\{s^{(j,1)}\}_{j\in[J]} = [1 \quad 0 \quad 0], \{s^{(j,2)}\}_{j\in[J]} = [2 \quad 0 \quad 0].
\end{aligned} \tag{9}$$

The resulting reward is

$$\begin{aligned}
\text{OPT}_W &= \sum_{t\in[T]} (C^t)' s^{([J],t)} - 0.1 \sum_{k\in[K]} f^j\left(s^{(j,t)}\right) \\
&= 2 - 0.1 \times \left( \left| 1 - \frac{1}{3} \right| + \left| 2 - \frac{2}{3} \right| + 2 \times \left( \frac{1}{3} + \frac{2}{3} \right) \right).
\end{aligned}$$

**Limited Feedback:** $C^t, w^{(j,t)}, P$ are unknown. What we could use is the sequence of limited feedback of some rewards associated with action vertices in $\mathrm{N}_{\text{out}}^{\text{t}}(a^i)$, where $a^{(i,t)} > 0$, according to $\mathcal{G}^t$. In this case, Figure 1(c) gives a time-invariant example, which needs to be disclosed before the first round. According to the definition of edges and $(a^2 \to a^3 \in E^t)$, if we set $a^{(2,t)} = 1$, the reward $r^t\left(\{a^{(i,t)}\}_{i\in[I]} = [0,1,0]\right)$ that we actually achieved will be disclosed immediately. At the same time, we will also observe the reward $r^t\left(\{a^{(i,t)}\}_{i\in[I]} = [0,0,1]\right)$ that we could have achieved if we selected action vertex $a^3$.

Finally, later in the paper (Case II), we will consider the case where the compatibility graph $\mathcal{G}^t$ is time-varying and unknown until the beginning of round $t$. This could model the situation, where e.g. a malfunction of certain action vertices.

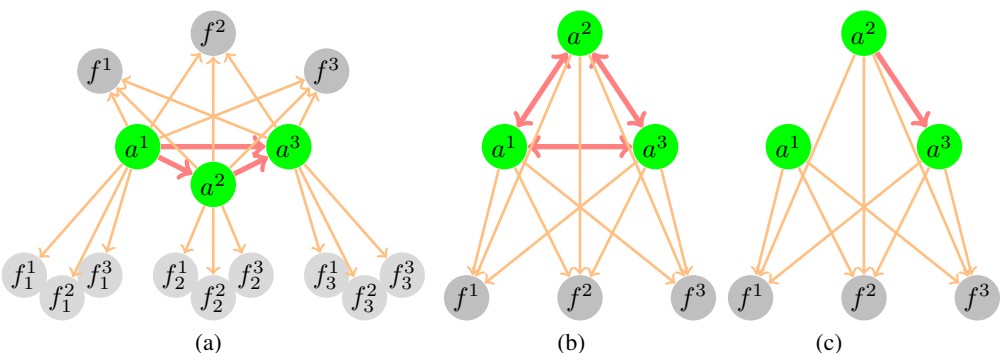

Figure 1: Examples of the compatibility graph with 3 action vertices (green), i.e., $a^1, a^2, a^3$ and 3 overall regulariser vertices (dark grey), i.e., $f^1, f^2, f^3$. Additionally, (a) has extra subgroup regulariser vertices (light grey), i.e., $f_k^1, f_k^2, f_k^3$ for each subgroup $k, k \in [1, 2, 3]$.

## 5 A MODEL WITH GRAPH-STRUCTURED BANDIT FEEDBACK

Let us consider a non-trivial special case of the general model, which is based on the large body of work within online convex optimisation with bandit feedback. Starting with Mannor & Shamir (2011), this line of work introduces a graphical feedback system to model side information available within online learning. In the graph, a directed edge starts vertex $a^i$ and ends vertex $a^j$ implies that by choosing vertex $a^i$ (i.e., $a^{(i,t)} = 1$) the reward associated with both vertices $a^i, a^j$ will be revealed immediately. Note that an in-directed edge between vertex $a^i$ and vertex $j^j$ can be seen as two directed edge $a^i \to a^j$ and $a^j \to a^i$. Thus, this line of work offers a middle ground between the bandits setting and the expert setting. [3]

This model applies particularly well to a variety of applications with partial observability of rewards. Following Mannor & Shamir (2011), there appeared an extensive literature incl. a wealth of beautiful results Alon et al. (2013; 2015; 2017). See Table 1 for a quick overview. In terms of multiple fairness regularisers, we model the compatibility graph as an directed, informed and non-stochastic variant in Alon et al. (2017).

### 5.1 CASE I: TIME-INVARIANT GRAPH-STRUCTURED BANDIT FEEDBACK

There are a number of algorithms available in the literature. Within the graph-structured feedback, Mannor & Shamir (2011) considered the ExpBan and ELP algorithms, out of which ELP allows for time-varying graph. Alon et al. (2015) devise bounds for three cases: strongly observable graphs, weakly observable graphs, and unobservable graphs. Starting with Mannor & Shamir (2011), their performance has been analysed such that the bounds are parameterised by structural parameters of the graphs. See Table 1 for an overview.

In this setting, reward function $r^t$ in round $t$ can be an arbitrary function of the history of actions, thus $c^t, w^{(j,t)}, P$ can be time-variant, unknown, and their explicit values will not be given after all rounds. The compatibility graph can be time-variant but disclosed in the beginning of each round. In round $t$, right after actions $a^{(i,t)} = 1$ is decided, not only the reward associated with the action vertex $a^i$, i.e., $r^t \left( a^{(i,t)} = 1; a^{(i^\dagger, t)} = 0, i^\dagger \in [I] \setminus \{i\} \right)$, but also the rewards associated with its out-neighbours $\mathrm{N}_{\mathrm{out}}^t(a^i)$ will be disclosed.

Let $q^{(i,t)}$ be the probability of observing the reward of action vertex $a^i$ in round $t$, $p^{(i,t)}$ be the probability of conducting action $a^i$ in round $t$ ($a^{(i,t)} = 1$). In Algorithm 1, $p^{(i,t)}$ is the trade-off

---

[3]In the bandits setting, only the loss of the chosen vertices is revealed and the best possible regret of order $\sqrt{\log(K)T}$, achieved by the Hedge algorithm Freund & Schapire (1997) or Follow the Perturbed Leader algorithm Kalai & Vempala (2005). On the other hand, the loss of all vertices is revealed in experts setting and the INF algorithm Audibert et al. (2009) achieves the optimal regret of order $\sqrt{KT}$.

---

**Algorithm 1** Exponentially-Weighted Algorithm with Linear Programming for the Model with Graph-Structured Feedback

---

**Input**: The number of action vertices $I$, regulariser vertices $J$ and rounds $T$, compatibility graph $\mathcal{G}$, confidence parameter $\delta \in (0, 1)$, learning rate $\eta \in (0, 1/(3I)]$.

**Output**: Action and states.

**Initialisation**: $\phi^{([I],1)} = 1, t = 1$. Let $\Delta_I$ be the $I$-dimensional probability simplex, and $\xi^{[I]}$ be a solution to the linear program

$$\max_{\xi^{[I]} \in \Delta_I} \min_{i \in [I]} \sum_{n \in \mathrm{N_{out}}(a^i)} \xi^{(n)} \tag{10}$$

1: **while** $t \in [T]$ **do**
2:     Set $p^{(i,t)} := (1-\gamma^t)\phi^{(i,t)}/\Phi^t + \gamma^t \xi^{(i)}$, where $\Phi^t = \sum_{i \in [I]} \phi^{(i,t)}$, $\gamma^t = \frac{(1+\beta)\eta}{\min_{i \in [I]} \sum_{n \in \mathrm{N_{out}}(a^i)} \xi^{(n)}}$
    and $\beta = 2\eta \sqrt{\frac{\ln(5I/\delta)}{\ln I}}$.
3:     Update $q^{(i,t)} = \sum_{i \in N(k)} p^{(i,t)}$.
4:     Draw one action vertex $a^i$ according to distribution $p^{([I],t)}$. Set

$$a^{(i,t)} = 1; \ a^{(i^\dagger,t)} = 0, \forall i^\dagger \in [I] \setminus \{i\}. \tag{11}$$

5:     Observe pairs $(n, r^{(n,t)})$ for all $n \in \mathrm{N_{out}}(a^i)$, in equation 12.

$$r^{(n,t)} = r^t \left( a^{(n,t)} = 1; a^{(i^\dagger,t)} = 0, i^\dagger \in [I] \setminus \{n\} \right), \forall n \in \mathrm{N_{out}}(a^i). \tag{12}$$

6:     For any $n \in [I]$, set estimated reward $\hat{r}^{(n,t)}$ and update $\phi^{(n,t+1)}$, as follows

$$\hat{r}^{(n,t)} = \frac{r^{(n,t)}\mathbb{I}\{n \in \mathrm{N_{out}}(a^i)\} + \beta}{q^{(n,t)}}, \tag{13}$$

$$\phi^{(n,t+1)} = \phi^{(n,t)} \exp(\eta \ \hat{r}^{(n,t)}) \tag{14}$$

7:     $t = t + 1$.
8: **end while**

---

between weight $\phi^{(i,t)}$ and $\xi^{(i)}$, with an egalitarianism factor $\gamma^t$, where $\phi^{(i,t)}$ is the weight for each action vertex. The relevant weight of an action vertex increases (decreases) when a payoff is good (bad). $\xi^{(i)}$ in equation 10 represent the desire to pick an action vertex uniformly from each clique among action vertices in $\mathcal{G}^t$. Then, we can update the importance sampling estimator $\hat{r}^{(i,t)}$ and the weight $\phi^{(i,t)}$ for $i \in [I]$, as in (13-14).

We can bound the actual regret with respect to the best single action in hindsight, with probability at least $1 - \delta$ with respective to the user's internal randomisation. In in the interest of space, we present the results that are a special case of Theorem 1 below only in the Supplementary Material.

## 5.2 CASE II: TIME-VARYING GRAPH-STRUCTURED BANDIT FEEDBACK

Finally, let us consider rather a general model, which extends the work on online convex optimisation with bandit feedback to time-varying graphs $\mathcal{G}^t$ describing the feedback. We present the Algorithm 2 for this case in the Supplementary Material. We also show:

**Theorem 1** (Informal version). *Algorithm 1 achieves weak regret of $\tilde{\mathcal{O}}\left(\sqrt{\log(I/\delta) \sum_{t \in [T]} mas(\mathcal{G}_t)}\right)$, where $\tilde{\mathcal{O}}$ hides only numerical constants and factors logarithmic in the number of actions $I$ and $1/\eta$ and $mas(\mathcal{G}_t)$ is the size of the maximal acyclic subgraph in $\mathcal{G}_t$.*

The formal version of the theorem and its proof are included in the Supplementary Material, but essentially follow the work of Alon et al. (2017).

## 6 A NUMERICAL ILLUSTRATION

Let us revisit the second Motivating Example of p. 4 and consider two major political parties, i.e., Conservative and Liberal parties, and several third-party candidates ($J = 3$). The states $\left\{s^{(j,t)}\right\}_{j\in[J]}$ represent how much of the advertising space-time is allocated to each political party $j \in [J]$ in round $t \in [T]$. To make the example more interesting, consider three equally-weighted regularisers that require the two major parties and all third-party candidates to get a $0.3, 0.6, 0.1$ share of advertising space-time on social networking sites respectively, e.g., to match the most recent election results. Then, $s^{([J],t)}/\sum_{j\in[J]} s^{(j,t)} = [0.3, 0.6, 0.1]$. The three regularisers are all weighted by $w^{(j,t)} = 0.1$, for $j \in [J], t \in [T]$. Each regulariser returns a penalty, i.e., a minus reward, for the state of its corresponding party (parties) being away from the target share, such that $f^j(s^{(j,t)}) = |s^{(j,t)} - \frac{s^{(j,t)}}{\sum_{j\in[J]} s^{(j,t)}}|$. The revenue of the platform, is assumed to be $c^t(s^{([J],t)}) := (C^t)'s^{([J],t)}$, where $C^t \in \mathbb{R}^J$ is the revenue vector. We test our method on the revenue of 3 advertisers and $100,000$ impressions (a $3 \times 100,000$ revenue matrix) from the dataset of Lu et al. (2020). Each row of the revenue matrix represents the revenue vector $C^t$ of one round.

We consider three action vertices $a^1, a^2, a^3$ ($I = 3$) as selling one unit of advertising space-time to Conservative party, Liberal party, and third-party candidates. Further, We set the state transmission function $s^{(n,t)} = s^{(n,t-1)} + a^{(n,t)}$, and the initial states for all parties $s^{([J],0)} = [0, 0, 0]$. For instance, if we conduct action $a^{(1,t)} = 1$, then $s^{(1,t)} = s^{(1,t-1)} + 1$ but $s^{(j,t)} = s^{(j,t-1)}$ for $j = 2, 3$. In this case, although only the state $s^{(1,t)}$ changes, the penalties from the all regularisers $f^2(s^{(2,t)}), f^3(s^{(3,t)})$ change correspondingly. Therefore, there must be directed edges from each action vertex to all regulariser vertices in the compatibility graph, such as Figure 1(b) and 1(c).

For Algorithm 1 and 2, the values of learning rate and confidence parameter $\eta, \delta$ are randomly chosen from the range of $[0, 1/3I]$ and $[0.1/4]$, respectively. A compatibility graph, with maximum acyclic subgraph less than 2 among all action vertices, is randomly generated in each trial for Algorithm 1, and in each round for Algorithm 2. Let the length of time window be chosen from the range of $[30, 70]$ with a gap of 5. For each time window ($T$), 5 trials are conducted and each trial randomly takes $T$ rows of the revenue matrix. We illustrate the dynamic (blue) and weak (green) regrets in Figure 2. The objective values (rewards) of $\text{OPT}_D, \text{OPT}_W$, solved via CVXPY library in three cases, and $R(A_1), R(A_2)$ are shown in Figure 3. In both figures, the mean and mean $\pm$ one standard deviations across 5 trials are presented by curves with shaded error bands. Our implementation is in Supplementary Material and will be made public once accepted.

## 7 CONCLUSIONS

We have introduced a general framework integrating feedback from multiple fairness measures, some of which may be conflicting, and whose weights and relations among actions and other measures may change over time.

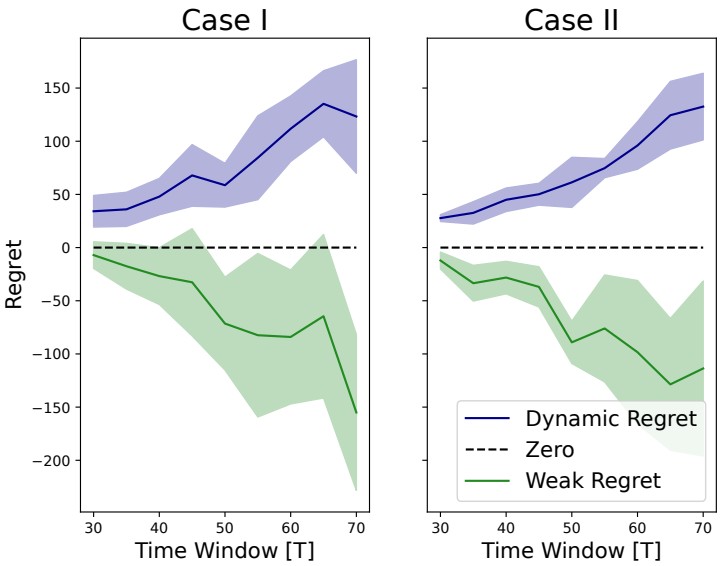

Figure 2: Dynamic (blue) and weak (green) regrets of Algorithm 1-2 implemented for 5 trials, with a randomly selected batch of revenue vectors for each trial. The mean regret and mean $\pm$ one standard deviations across 5 trials are presented by curves with shaded error bands.

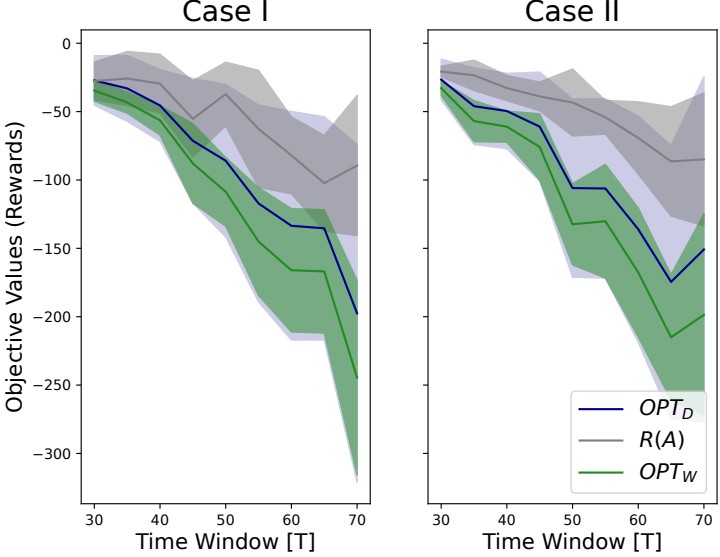

Figure 3: Rewards of Algorithm 1-2 (grey) and objective values of $\text{OPT}_D$ (blue), $\text{OPT}_W$ (green) implemented for 5 trials, with a randomly selected batch of revenue vectors for each trial. The mean reward and mean $\pm$ one standard deviation across 5 trials are presented by curves with shaded error bands.

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

# 8 APPENDIX

## 8.1 RELATED WORK

There are many measures of fairness for any single subgroup and many protected attributes (e.g., gender, race, ethnicity, income) defining the subgroups. Without attempting to encompass all of the related literature, let us present some of the most relevant work.

### 8.1.1 THE CONFLICT BETWEEN GROUP AND INDIVIDUAL FAIRNESS

Especially in the USA, affirmative action policies are often controversial because favouring one group inevitably involves disadvantaging another Dur et al. (2020). In the suitcase Regents of the University of California v. Bakke (1978), race is allowed to be one of several factors in college admission policy. California Proposition 209 prohibits states governmental institutions from considering race, sex, or ethnicity, specifically in the areas of public employment, public contracting, and public education. In 2019, the California Senate Constitutional Amendment No. 5 (SCA-5) asks to eliminate California Proposition 209's ban on the use of race, sex, etc. Asian Americans opposed this amendment Wang (2020). Behind those debates is the collision of different fairness perceptions originating from distinct view on which factors of their performance individuals should be held accountable for Schildberg-Hörisch et al. (2020).

### 8.1.2 MEASURES OF FAIRNESS

There are many definitions of fairness, esp. within the applications of fairness to classification. The statistical definition of fairness is to request a classifier's statistics, such as raw positive classification rate (also sometimes known as statistical parity), false positive, and false negative rates (also sometimes known as equalised odds), be equalised across the subgroups so that the error caused by the algorithm be proportionately spread across subgroups. "Demographic parity", proposed by Calder et al. (2009), requires the proportion of each segment of a protected class (e.g. gender) should receive the positive outcome at equal rates. However, it might be unfair in the case of unbalanced distributions of features between advantaged and disadvantaged subgroups even in the absent of biases. The notions of "equal odd" in Hardt et al. (2016) and "counterfactual fairness " of Kusner et al. (2017) require the predictor to be unrelated to protected attributes. In other words, they can be seen as the revised version of unawareness that gets rid of the effects of those unobserved features related to protected attributes. They focus more on accurate prediction of the unbalanced distribution without discrimination. It believes that a predictor is very unlikely to be discriminatory if it only reflects the real outcomes.

Group fairness only provides an average guarantee for the individuals in a protected group Awasthi et al. (2020) and is insufficient by itself. Sometimes even the notions of group fairness is maintained, from the view of an individual, the outcome is unfair. The individual definition asks for constraints that bind on specific pairs of individuals, rather than on a quantity that is averaged over groups or in other words, it requires "similar individuals should be treated similarly" Dwork et al. (2011). This notion requires a similarity metric capturing the ground truth, which requires general and task-specific agreement on its definition.

### 8.1.3 FRAMEWORKS FOR REASONING ABOUT MULTIPLE FAIRNESS MEASURES

Very recently, there have been several attempts at formulating frameworks for reasoning about multiple fairness measures. The fairness resolution model, proposed in Awasthi et al. (2020), is guided by the unfairness complaints received by the system and it can be a more practical way to maintain both group and individual fairness. This work provides a finite-state discrete-time Markov chains framework that supports multiple fairness regularisers and takes into account their potential incompatibilities. It leads, however, to PSPACE-Complete decision problems, which are hard to solve in practice, independent of whether P equals NP. Independently, Ospina et al. (2021) proposed a framework that utilizes an online algorithm for time-varying networked system optimisation, with the aim to tradeoff human preferences. Particularly, the function of human preference (fairness regulariser function) are learned concurrently with the execution, using shape-constrained Gaussian processes.

Alternatively, one could consider approaches from multi-objective optimisation (MOO), where key recent references include Zhang et al. (2021) who consider dynamic MOO, but do not provide any guarantees on the performance of the algorithms. This, in turn, is based on a long history of work on convexifications (Sun et al., 2001, e.g.) in MOO. There is also related work across many applications of MOO, such as in Mathematical Finance Li & Ng (2000).

### 8.1.4 TECHNICAL TOOLS

Once we formulate our framework, we present algorithms applicable therein, which draw upon the work on online non-smooth convex optimisation Zinkevich (2003). We refer to Cesa-Bianchi & Lugosi (2006); Shalev-Shwartz et al. (2011); Hazan (2016) for recent book-length introductions. In the case of graph-structured feedback, there are algorithms summarised in Table 1, which are based on the tradition of online convex optimisation with bandit-feedback Kleinberg (2004); Agarwal et al. (2010); Bubeck et al. (2015); Hazan & Li (2016); Ito (2020). We discuss these in Table 1.

## 8.2 ALGORITHM OF CASE II

In the setting, the compatibility graph can be time-variant. The Algorithm 2 for case II uses the similar method as case I but needs to solve a linear program inequation 15 whenever a new graph comes.

## 8.3 PROOF OF THEOREM 1

*Proof.* We want to show that with learning rate $\eta \leq 1/(3K)$ sufficiently small such that $\beta \leq 1/4$, with probability at least $1 - \delta$, we have that $\text{Regret}(A_2)$ is upper bounded by equation 20, where $\tilde{\mathcal{O}}$ hides only numerical constants and factors logarithmic in K and $1/\eta$.

$$\sqrt{5\log(\frac{5}{\delta})\sum_{t\in[T]}\text{mas}(\mathcal{G}_t)}+12\eta\sqrt{\frac{\log(5K/\eta)}{\log K}\sum_{t\in[T]}\text{mas}(\mathcal{G}_t)} \qquad (20)$$
$$+\tilde{\mathcal{O}}\left(1+\sqrt{T}\eta+T\eta^2\right)\left(\max_{t\in[T]}\text{mas}^2(\mathcal{G}_t)\right)+\frac{2\log(5K/\delta)}{\eta},$$

---

**Algorithm 2** Exponentially-Weighted Algorithm with Linear Programming for the Model with Graph-Structured Feedback

---

**Input**: The number of action vertices $I$, regulariser vertices $J$ and rounds $T$, compatibility graph $\mathcal{G}^t$ in each round, confidence parameter $\delta \in (0, 1)$, learning rate $\eta \in (0, 1/(3I)]$.
**Output**: Actions and states.
**Initialisation**: $\phi^{([I],1)} = 1, t = 1$.

1: **while** $t \in [T]$ **do**
2:     Receive compatibility graph $\mathcal{G}^t$, out-neighbours $\mathrm{N}^t_{\mathrm{out}}(a^i), i \in [I]$ and in-neighbours $\mathrm{N}^t_{\mathrm{in}}(f^j), j \in [J]$.
3:     Let $\Delta_I$ be the I-dimensional probability simplex, and $\xi^{[I]}$ be a solution to the linear program

$$\max_{\xi^{[I]} \in \Delta_I} \min_{i \in [I]} \sum_{n \in \mathrm{N}_{\mathrm{out}}(a^i)} \xi^{(n)} \tag{15}$$

4:     Set $p^{(i,t)} := (1-\gamma^t)\phi^{(i,t)}/\Phi^t + \gamma^t \xi^{(i)}$, where $\Phi^t = \sum_{i \in [I]} \phi^{(i,t)}$, $\gamma^t = \frac{(1+\beta)\eta}{\min_{i \in [I]} \sum_{n \in \mathrm{N}_{\mathrm{out}}(a^i)} \xi^{(n)}}$
    and $\beta = 2\eta\sqrt{\frac{\ln(5I/\delta)}{\ln I}}$. Update $q^{(i,t)} = \sum_{i \in N(k)} p^{(i,t)}$.
5:     Draw one action vertex $a^i$ according to distribution $p^{([I],t)}$. Set

$$a^{(i,t)} = 1; \ a^{(i^\dagger,t)} = 0, \forall i^\dagger \in [I] \setminus \{i\}. \tag{16}$$

6:     Observe pairs $(n, r^{(n,t)})$ for all $n \in \mathrm{N}_{\mathrm{out}}(a^i)$, where

$$r^{(n,t)} = r^t \left( a^{(n,t)} = 1; a^{(i^\dagger,t)} = 0, i^\dagger \in [I] \setminus \{n\} \right), \forall n \in \mathrm{N}_{\mathrm{out}}(a^i). \tag{17}$$

7:     For any $n \in [I]$, set estimated reward $\hat{r}^{(n,t)}$ and update $\phi^{(n,t+1)}$, as follows

$$\hat{r}^{(n,t)} = \frac{r^{(n,t)}\mathbb{I}\{n \in \mathrm{N}_{\mathrm{out}}(a^i)\} + \beta}{q^{(n,t)}}, \tag{18}$$

$$\phi^{(n,t+1)} = \phi^{(n,t)} \exp(\eta \, \hat{r}^{(n,t)}) \tag{19}$$

8:     $t = t + 1$.
9: **end while**

---

To prove this, we refer to Theorem 9 in Alon et al. (2017). (21-22) use the definition of $\Phi^t, \phi^{(i,t)}, p^{(k,t)}$ in Algorithm 1. equation 23 uses inequality $\exp(x) \leq 1 + x + x^2$.

$$\frac{\Phi^{t+1}}{\Phi^t} = \sum_{k \in [K]} \frac{\phi^{(k,t+1)}}{\Phi^t} = \sum_{k \in [K]} \frac{\phi^{(i,t)}}{\Phi^t} \exp(\eta \hat{r}^{(k,t)}) \tag{21}$$

$$= \sum_{k \in [K]} \frac{p^{(k,t)} - \gamma^t \xi^{(i,t)}}{1 - \gamma^t} \exp(\eta \hat{r}^{(k,t)}) \tag{22}$$

$$\leq \sum_{k \in [K]} \frac{p^{(k,t)} - \gamma^t \xi^{(i,t)}}{1 - \gamma^t} \left( 1 + \eta \hat{r}^{(k,t)} + (\eta \hat{r}^{(k,t)})^2 \right) \tag{23}$$

$$\leq 1 + \frac{\eta}{1 - \gamma^t} \sum_{k \in [K]} \left( p^{(k,t)} \hat{r}^{(k,t)} + \eta p^{(k,t)} (\hat{r}^{(k,t)})^2 \right). \tag{24}$$

equation 25 uses equation 24 and inequality $\ln(x) \leq x - 1$.

$$\ln \left( \frac{\Phi^{T+1}}{\Phi^1} \right) = \sum_{t \in [T]} \ln \left( \frac{\Phi^{t+1}}{\Phi^t} \right) \leq$$
$$\sum_{t \in [T]} \sum_{k \in [K]} \frac{\eta}{1 - \gamma^t} \left( p^{(k,t)} \hat{r}^{(k,t)} + \eta p^{(k,t)} (\hat{r}^{(k,t)})^2 \right). \tag{25}$$

For a fixed single action on vertex $v^k$, we have

$$
\ln\left(\frac{\Phi^{T+1}}{\Phi^1}\right) \geq \ln\left(\frac{\phi^{(k,T+1)}}{\Phi^1}\right) =
$$
$$
\ln\left(\frac{\phi^{(k,1)}\exp(\eta\sum_{t\in[T]}\hat{r}^{(k,t)})}{K}\right) = \eta\sum_{t\in[T]}\hat{r}^{(k,t)} - \ln K. \tag{26}
$$

From Azuma's inequality, Chernoff bound and Freedman's inequality, we have the upper bound of regret with probability at least $1 - \delta$ in equation 27. Then, equation 28 is obtained by combining (25-26). By substituting condition $\beta = \tilde{\mathcal{O}}(\eta), \gamma^t = \tilde{\mathcal{O}}(\eta\text{mas}(\mathcal{G}^t)) \in [\eta, 1/2]$, we get the upper bound in equation 20, where $\tilde{\mathcal{O}}$ ignores factors depending logarithmically on K and $1/\delta$.

$$
\sum_{t\in[T]} r^{(k,t)} - r^{(\mathcal{I}^t,t)}
$$
$$
\leq \left(\sum_{t\in[T]}\hat{r}^{(k,t)} - \sum_{t\in[T]}\sum_{k\in[K]}p^{(k,t)}\hat{r}^{(k,t)}\right) + \frac{\ln(k/\delta)}{\beta} + \sqrt{\frac{T\ln(K/\delta)}{2}} \tag{27}
$$
$$
+ \sqrt{2\ln(1/\delta)\sum_{t\in[T]}\text{mas}(\mathcal{G}^t)} + \beta\sum_{t\in[T]}\text{mas}(\mathcal{G}^t) + \tilde{\mathcal{O}}\left(\max_{t\in[T]}\text{mas}(\mathcal{G}^t)\right)
$$
$$
\leq \frac{\eta}{1 - \max_{t\in[T]}\gamma^t}\sum_{t\in[T]}\sum_{k\in[K]}\left(p^{(k,t)}(\hat{r}^{(k,t)})^2 + \gamma^t p^{(k,t)}\hat{r}^{(k,t)}\right) + \frac{\ln K}{\eta} \tag{28}
$$
$$
+ \frac{\ln(k/\delta)}{\beta} + \sqrt{\frac{T\ln(K/\delta)}{2}} + \sqrt{2\ln(1/\delta)\sum_{t\in[T]}\text{mas}(\mathcal{G}^t)} + \beta\sum_{t\in[T]}\text{mas}(\mathcal{G}^t) + \tilde{\mathcal{O}}\left(\max_{t\in[T]}\text{mas}(\mathcal{G}^t)\right).
$$

$\square$

## 8.4 DETAILS OF THE NUMERIC ILLUSTRATION

We represent the experimental results associated with Figure 3 in another format, as in Figure **??**. Note that the problem of $\text{OPT}_D, \text{OPT}_W$ in three cases are the same because we are using the same dataset. Therefore, we illustrate the objective values (rewards) of $\text{OPT}_D$ across three cases as purple and those of $\text{OPT}_W$ as pink. The rewards from Algorithm **??**-3 are presented as blue, green and yellow respectively. Further, the bubble displays the reward across 5 trials against the length of time window, with its centre showing the mean reward and radius showing one standard derivation.

| Symbol | Meaning |
|---|---|
| $T$ | the number of rounds. |
| $I$ | the number of action vertices. |
| $J$ | the number of regulariser vertices. |
| $[T], [0, T-1]$ | $[T] = 1, \ldots, T$ and $[0, T-1] = 0, 1, \ldots, T-1$. |
| $\mathcal{G}, \mathcal{G}^t$ | time-invariant and time-variant compatibility graphs. |
| $a^i$ | an action vertex. |
| $f^j$ | a regulariser vertex. |
| $V_a$ | the set of action vertices, and $V_a = \{a^1, a^2, \ldots, a^I\}$. |
| $V_f$ | the set of regulariser vertices, and $V_f = \{f^1, f^2, \ldots, f^J\}$. |
| $a^{i1} \to a^{i2}$ | a directed edge from $a^{i1}$ to $a^{i2}$. If action $a^{i1}$ is conducted, the reward associated with $a^{i2}$ will be disclosed together with $a^{i1}$. |
| $a^i \to f^j$ | a directed edge from $a^i$ to $f^j$. It means that conducting action $a^i$ can affect regulariser $f^j$. |
| $N_{out}^t(a^i)$ | the out-neighbours of action vertex $a^i$, defined in equation 1. |
| $N_{in}^t(f^j)$ | the in-neighbours of regulariser vertex $f^j$, defined in equation 2. |
| $E, E^t$ | the set of time-invariant and time-variant (un)directed edges. |
| $a^{(i,t)}$ | the indicator of choosing action vertex $a^i$ in round $t$ ($a^{(i,t)} = 1$) or not ($a^{(i,t)} = 0$). |
| $s^{(j,t)}$ | the state of regulariser vertex $f^j$ in round $t$ |
| $s^{([J],t)}, s^{(j,[T])}$ | the vector of $[s^{(1,t)}, \ldots, s^{(J,t)}]$ and $[s^{(j,1)}, \ldots, s^{(j,T)}]$ respectively. |
| $P$ | the state transmission function, defined in equation 3. |
| $r^t(a^{(i,t)}, i \in [I])$ | the reward obtained in round $t$. |
| $c(s^{(j,t)}, j \in [J])$ | the function of performance criterion, e.g., revenue. |
| $C^t$ | the revenue vector in round $t$. |
| $w^{(j,t)}$ | the weight of regulariser $f^j$ in round $t$. |
| $A, A_1, A_2$ | an algorithm. |
| $R(A)$ | the cumulative reward of the Algorithm A. $R(A) := \sum_{t \in [T]} r^t(a^{(i,t)}, i \in [I])$, where $\{a^{(i,t)}\}_{i \in [I]}$ is decided by Algorithm A. |
| $OPT_D$ | the best sequence of actions chosen in hindsight, defined in equation 5. |
| $OPT_W$ | the best single action chosen in hindsight, defined in equation 6. |
| Algorithms | |
| $q^{(i,t)}$ | the probability of observing the reward associated with action vertex $a^i$ in round $t$. |
| $p^{(i,t)}$ | the probability of selecting action vertex $a^i$ in round $t$. |
| $\phi^{(i,t)}$ | the weight for each action vertex in Algorithm 1 and 2. It increases (decreases) when a payoff is good (bad). |
| $\xi^{(i,t)}$ | the rates that represent the desire to pick an vertex uniformly from each clique in $\mathcal{G}$ or $\mathcal{G}^t$, defined in equation 10. |
| $\gamma^t$ | an egalitarianism factor controlling the trade-off between weight $\phi^{(i,t)}$ and $\xi^{(i,t)}$. |
| $\delta$ | the confidence parameter in Algorithm 1 and 2. |
| $\eta$ | the learning rate in Algorithm 1 and 2. |
| $\hat{r}^{(n,t)}$ | the importance sampling estimator of rewards, as in equation 13. |

Table 2: An overview of the notation

