# OpenReview forum: "Learning Time-Varying Convexifications of Multiple Fairness Measures"
_ICLR.cc/2024/Conference — Submitted to ICLR 2024_

### Official Review · Reviewer_7nxQ · 2023-10-29

**Soundness:** 3 good
**Presentation:** 3 good
**Contribution:** 2 fair
**Rating:** 6
**Confidence:** 3

**Summary:**

This paper proposes a new framework for integrating multiple and potentially evolving fairness criteria via graph networks. The relative weights of the fairness criterion are unknown a priori, and might change over time. The paper then studies algorithms which learn and integrate multiple fairness objectives under limited graph-structured feedback under two cases: (1) the graph structure is revealed and time-invariant (2) the graph structure is revealed at the beginning of each round, and might change over time. Theoretical upper bound on the "weak regret" (i.e. regret w.r.t the single best action in hindsight) is provided for the two cases. The paper also provides numerical results with an advertisement dataset for the weak regret, and the dynamic regret (regret w.r.t. the best sequence of actions in hindsight).

**Strengths:**

- The paper studies a significant and practical issue about integrating feedbacks from multiple fairness measures. In particular, these fairness measures could be conflicting to each other, or weighted differently at different times. I found the partial graph feedback framework proposed by the authors to be interesting and reasonable.
- Overall the paper has a good clarity (except some typos, see below). The illustrative example was helpful in understanding how the OPT might look like and how the general framework might fit into use cases in practice.
- The paper provides a good combination of theoretical analysis and numerical experiments, and the results demonstrate the difficulty of bounding the dynamic regret.

**Weaknesses:**

- The exponentially-weighted algorithm only provides an upper bound for the weak regret, and there is few analysis / discussions on the optimality of such bound, and any upper / lower bounds analysis for the dynamic regret.
- The numerical evaluations are on a relatively weak side. The fairness measures from the regularisers in the experiments are rather simple where each regulariser penalize the deviation from a fixed target share (essentially the same type of fairness measure). The results can be strengthened by demonstrating with multiple different fairness measures.
- In the example in Section 4, the fairness reward from the regulariser f^j at round t is not only a function of its own state, but also a function of all other regularisers' states at round t. I'd suggest to change $f^j(s^{(j, t)})$ in eq(4) to include other regularisers' states.

typos:
- eq(5) and eq(6), extra "$\vert$" under "max"
- paragraph above eq(6), extra $\in$

**Questions:**

- are there any assumptions on the functional properties of the state evolving function P?

---

### Official Review · Reviewer_wCNF · 2023-10-30

**Soundness:** 2 fair
**Presentation:** 1 poor
**Contribution:** 2 fair
**Rating:** 3
**Confidence:** 2

**Summary:**

The paper addresses a critical case, that in many scenarios, multiple fairness measures may be important to consider. Even more subtle, the relative importances of those measures are not previously known and therefore must be learned during the process. Such time-varying factor causes the learning process more complicated. This paper aims to build a protocol for this purpose with some help from limited graph-structured feedback.

**Strengths:**

The paper addresses an interesting topic and Table 1 provides a comprehensive overview for earlier algorithms, which is helpful for readers who want to investigate further on this topic.

Chapter 4 is an interesting read: motivating real-life examples are highly relevant with this topic (fairness) overall and may stimulate readers' interests to a great extent.

**Weaknesses:**

Graph-structured feedback is a fairly uncommon concept in the ML literature. Therefore, it would be immensely helpful if the paper has a more comprehensive section focusing on the preliminary materials. For example, what really are the "action vertices"? Does "action" refer to the concept in reinforcement learning, and even if so, what is its relationship with reducing unfairness here?

(This is relevant with the previous comment above) Many equations are unclearly written and explained. Equation 3 is an example: what is a precise definition of a state evolution function $P$, specifically over the case here? Many other equations in this paper shall be improved on this aspect.

The general setting (time-varying compatibility graph and unknown relative weights) and the reasons to apply graph-structured feedback may need to be further justified. In particular, why is such a graph structure with those definitions a good fit for this task? In many scenarios, the fairness criterion that need to be preserved are generally already known (e.g. income, race, etc), and would learning such weights may even add noises?

**Questions:**

Please refer the third paragraph on the Weaknesses section.

---

### Official Review · Reviewer_3uXb · 2023-11-01

**Soundness:** 2 fair
**Presentation:** 2 fair
**Contribution:** 2 fair
**Rating:** 3
**Confidence:** 3

**Summary:**

This paper proposes a general framework for integrating conflicting fairness measures into decision-making algorithms, recognizing the increasing importance of balancing them. To achieve this, the authors employs established online learning algorithms using graph-structured feedback (Mannor \& Shamir, 2011), representing the relations between each fairness regularizer as a graph. Notably, this paper addresses both time-invariant and time-varying cases, accounting for potential shifts in the relative importance of fairness regularizers over time.

**Strengths:**

As demonstrated in the advertising example presented in the paper, considering the balance between various fairness regularizers in decision-making is crucial for devising a fair decision-making algorithm. Additionally, the setting where the weights between regularizers change over time is also particularly intriguing.

**Weaknesses:**

Although the paper is understandable, the structure/organization of the presentation could be improved. Specifically, since this paper introduces a new framework, I believe that the problem formulation (Section 3) and motivating example (Section 4) are crucial for understanding the paper. However, there are some unclear definitions and inconsistencies in the example.

It is not clear as to what specific technical novelty exists beyond the proposal of a new framework. The description of how Algorithm 1 differs from ELP algorithm in Mannor \& Shamir (2011) is lacking, as well as an explanation of the challenges faced in regret analysis and how they were overcome.

**Questions:**

The authors have presented dynamic regret and weak regret as performance measures, but they have only provided theoretical results for weak regret. Are there any specific challenges in obtaining results for dynamic regret?

What is the definition of a concave combination in equation (4)? If it is meant to be a convex combination, i.e., $\sum_j w^{(j,t)} f^j (s^{(j,t)})$ where $w^{(j,t} \ge 0, \sum_j w^{(j,t)} = 1$, in the motivating example on page 5 where a single scalar weight $w^{(j,t)}=0.1$ is discussed, it seems that the fairness reward does not a convex combination.

In Figure 1-(a), how do subgroup regularizer vertices differ from regularizer vertices? According to the definition of the graph $\mathcal{G}^t$ in Section 3, since $a^1_1$ in Figure 1-(a) has a total of 6 edges, this means that $a^1_1$ affects $\{ f^1, f^2, f^3, f^1_1, f^2_1, f^3_1 \}$. (In contrast, in the main text, it was mentioned that $a^1$ could potentially affect $\{ f^1, f^1_1, f^2_1, f^3_1 \}$.)

[Minor typo]
In eq(4), $c^t \left( s^{(i,t)}, \ldots, s^{(J,t)} \right) \rightarrow c^t \left( s^{(1,t)}, \ldots, s^{(J,t)} \right)$

The references in equations (1) and (2) do not align with the content of the main text.

In equation (8), the index for summation in $\text{OPT}_D$ should be changed from $k$ to $j$.

---

### Meta-Review · Area_Chair_xfdp · 2023-12-12

**Metareview:**

The reviewers are not positive about the contribution of the paper, the authors also did not post a rebuttal. So we reject the submission.

**Justification For Why Not Higher Score:**

N/A

**Justification For Why Not Lower Score:**

N/A

---

### Decision · Program_Chairs · 2024-01-16

Reject